# Development of New Predictive Equations for the Resting Metabolic Rate (RMR) of Women with Lipedema

**DOI:** 10.3390/metabo14040235

**Published:** 2024-04-19

**Authors:** Małgorzata Jeziorek, Jakub Wronowicz, Łucja Janek, Krzysztof Kujawa, Andrzej Szuba

**Affiliations:** 1Department of Dietetics and Bromatology, Faculty of Pharmacy, Wroclaw Medical University, 50-367 Wroclaw, Poland; 2Statistical Analysis Center, Wroclaw Medical University, 50-372 Wroclaw, Poland; jakub.wronowicz@umw.edu.pl (J.W.); lucja.janek@umw.edu.pl (Ł.J.); krzysztof.kujawa@umw.edu.pl (K.K.); 3Department of Angiology and Internal Medicine, Wroclaw Medical University, 50-367 Wroclaw, Poland; andrzej.szuba@umw.edu.pl

**Keywords:** predictive equation, resting metabolic rate, lipedema

## Abstract

This study aimed to develop a novel predictive equation for calculating resting metabolic rate (RMR) in women with lipedema. We recruited 119 women diagnosed with lipedema from the Angiology Outpatient Clinic at Wroclaw Medical University, Poland. RMR was assessed using indirect calorimetry, while body composition and anthropometric measurements were conducted using standardized protocols. Due to multicollinearity among predictors, classical multiple regression was deemed inadequate for developing the new equation. Therefore, we employed machine learning techniques, utilizing principal component analysis (PCA) for dimensionality reduction and predictor selection. Regression models, including support vector regression (SVR), random forest regression (RFR), and k-nearest neighbor (kNN) were evaluated in Python’s scikit-learn framework, with hyperparameter tuning via GridSearchCV. Model performance was assessed through mean absolute percentage error (MAPE) and cross-validation, complemented by Bland–Altman plots for method comparison. A novel equation incorporating body composition parameters was developed, addressing a gap in accurate RMR prediction methods. By incorporating measurements of body circumference and body composition parameters alongside traditional predictors, the model’s accuracy was improved. The segmented regression model outperformed others, achieving an MAPE of 10.78%. The proposed predictive equation for RMR offers a practical tool for personalized treatment planning in patients with lipedema.

## 1. Introduction

Precisely estimating the energy needs of individuals has numerous valuable clinical applications. One evident application is in managing the weight of individuals, especially when they are dealing with obesity [1]. Lipedema is often misdiagnosed as obesity; however, the two frequently co-occur. The majority of women with lipedema are also overweight or have obesity [2,3]. Lipedema is a chronic progressive disease that primarily affects women. It is characterized by an excessive accumulation of fat on the legs (without involvement of the feet) and in one third of all patients, on the arms. The most common clinical symptoms of lipedema include a disproportion between the upper and lower parts of the body, spontaneous or palpation-induced pain, and easy bruising. The etiology of the disease is still unknown, but several factors may be involved, such as genetic predisposition, hormonal influence, changes in fat cells, microvascular dysfunction, capillary damage, lymphatic disturbances, and inflammation [4,5,6]. The progression of lipedema is linked with weight gain; therefore, weight management and obesity treatment are crucial factors in the treatment of lipedema [7,8,9]. Lipedema is frequently confused with lymphedema, a condition directly linked to lymphatic system failure or damage to lymphatic vessels. Table 1 provides a comparison to help distinguish between lipedema and lymphedema [2,5,6].

Resting metabolic rate (RMR) is a crucial component in estimating overall energy expenditure. RMR is the amount of energy, measured in calories, that the body requires at rest in order to maintain basic physiological functions such as breathing, circulation, and cell production. It represents the minimum amount of energy needed to sustain life when the body is in a completely rested and fasting state. The body’s metabolism is the process by which it converts food and nutrients into energy. RMR specifically refers to the energy expended during these basic bodily functions while at rest. It does not take into account the energy expended during physical activity or the thermic effect of food [10,11]. RMR typically constitutes the majority, ranging from 60% to 75%, of the total energy expenditure (TEE) [1]. Several factors influence an individual’s RMR, including body composition, especially fat-free mass, age, gender, genetics, hormones, ethnicity, physical fitness, and a range of related environmental factors, including diet [11,12].

Individuals with lipedema may have unique metabolic characteristics compared to the general population. In our previous study, we demonstrated that commonly used equations for normal or overweight/obese patients have limited applicability in individuals with lipedema. Our study revealed a low agreement of predictive equations compared to actual RMR measured by an indirect calorimetry (IC) in lipedema patients (less than 60%). The most frequently applied equations prove ineffective in clinical practice within this specific population, primarily due to substantial individual variations among the women studied [13]. We also concluded that IC is the most reliable tool for assessing RMR in patients with lipedema. However, given its limited availability and high cost, there was a necessity to propose a new equation for determining RMR in clinical practice when IC is not accessible (e.g., in a dietitian’s office, health clinic, or hospital). Accurate knowledge of RMR helps in calculating daily caloric needs, which is essential for weight management, designing appropriate nutrition plans, and avoiding under- or overestimation of energy requirements [14]. By having specific predictive equations for RMR in women with lipedema, healthcare professionals can tailor treatment plans, including dietary recommendations and exercise regimens, to better meet the metabolic needs of these individuals. Having reliable predictive equations for RMR in women with lipedema can enhance the precision of clinical assessments and research studies focused on metabolic aspects of lipedema. This can contribute to a deeper understanding of the condition and facilitate the development of more effective interventions. Healthcare providers can use accurate predictive equations for RMR to offer more personalized and effective care to women with lipedema. This in turn may lead to better adherence to treatment plans and improved long-term outcomes.

The aim of this study was to derive a predictive equation for resting metabolic rate (RMR) using a sample of 119 women with lipedema.

## 2. Materials and Methods

### 2.1. Study Population

The investigation included a total of 119 female participants recruited from the Angiology Outpatient Clinic at Wroclaw Medical University in Poland. This cohort comprised individuals diagnosed with lipedema based on typical clinical symptoms by an angiologist [15]. Exclusion criteria encompassed pregnancy, breastfeeding, a post-pregnancy period of 6 months, various medical conditions such as lymphedema, edema associated with chronic vein insufficiency or heart failure, diabetes mellitus, kidney or liver failure, hormonally imbalanced thyroid disease, cancer, and the presence of implanted devices. The research adhered to the Declaration of Helsinki and received approval from the Bioethics Committee at Wroclaw Medical University, Poland (KB-456/2019). All participants provided written informed consent. The study commenced in January 2020 and concluded in June 2021.

### 2.2. Body Composition and Anthropometry

The Tanita HR-001 growth meter (Tanita, Tokyo, Japan) was utilized for height assessment, while the Tanita MC-780MA (Tanita, Japan) was employed for measuring weight and various body composition parameters. Parameters including body fat percentage, body fat (kg), lean body mass (kg), total body water (kg), and visceral fat level were recorded. Participants were advised to abstain from food or drink for a minimum of 4 h, refrain from vigorous physical activity for 12 h, and avoid diuretics for 6 h prior to the study. Body mass index (BMI) was computed as the ratio of body weight (kg) to height (m) squared. Measurements of waist, hip, thigh, calf, and ankle circumferences were conducted using a standard tape measure, with accuracy to the nearest 1 cm.

### 2.3. RMR Measurement

Gas exchange was assessed using the FitMate WM device (Cosmed, Rome, Italy) through indirect calorimetry (IC). IC, recognized as the gold standard for measuring resting metabolic rate (RMR), involves quantifying oxygen consumption (VO_2_) and carbon dioxide production (VCO_2_). The respiratory quotient (RQ), represented by the ratio of VCO_2_ to VO_2_ (VCO_2_/VO_2_), determines the substrate utilization. The calorimeter collects breath exchange for gas analysis, allowing the determination of RMR using Weir’s equation (resting energy expenditure (REE) (kcal/day) = [(VO_2_·3.941) + (VCO_2_·1.11)]·1440) [16].

Prior to the measurement, participants were instructed to abstain from food and drinks (except water) for 8 h and vigorous exercise for 48 h. The assessment was conducted with participants in the supine position in a ventilated, dimly lit room maintained at a moderate temperature (22–26 °C) [17,18]. Participants were examined in the morning after 7–9 h of sleep. A 15 min rest in a seated position preceded the measurement to optimize conditions. Subsequently, participants donned a Fitmate WM face mask, and the measurement, lasting 10–20 min, was carried out in isolation to minimize external noise. Calibration of the Fitmate WM device was performed before each RMR assessment. The progress of the measurement was continuously monitored on the Fitmate WM screen. Participants were briefed on the procedural details before the study commenced.

### 2.4. Statistical Analysis

Similarly to other studies, we assumed the RMR to be linearly dependent on predictors. In such models, an assumption of non-multicollinearity of predictors should be met. Therefore, firstly multicollinearity was assessed using the variance inflation factor (VIF). We adopted a VIF > 10 as the criterion of multicollinearity presence. The VIF analysis revealed significant collinearity of all the predictors considered, as shown in Table 2.

As a result, classical multiple regression could not be reliably performed. Consequently, we resorted to employing machine learning (ML) methods instead. Multiple regression as a machine learning algorithm typically demands a sufficiently large sample (10 samples by predictor), and this requirement is contingent upon the number of explanatory variables. Given the insufficient data available (106 samples for 12 predictors) in our case, we opted for dimensionality reduction using principal component analysis (PCA). PCA is particularly effective when dealing with results linearly dependent on each variable and for standardized data. To ascertain linear dependence, we standardized the data by subtracting the mean and dividing by the standard deviation. The analysis revealed more or less linear dependence, prompting us to perform classical PCA without any kernel or rotation.

The determination of the number of principal components considered was guided by scree plot (Figure 1). We opted to retain the first three principal components, since collectively they accounted for 98.7% of the variance. This choice aligns with the outcome obtained using Kaiser’s rule (i.e., to retain all components with eigenvalues above 1.0).

To clarify how each variable is depicted within a specific component using the square cosine values (Cos2), we employed the “fviz_cos2” function from the fact extra package. The outcomes are visually represented in Figure 2, where the chart illustrates the square cosine (Cos2) values of variables concerning dimensions (Dim-1-2). A low value means that that variable is badly represented by that component, while a high value is connected to a good representation of the variable on that component [19]. The minimum value of a square cosine is 0 and the maximum 1.

Furthermore, we computed the quality of representation associated with each variable and incorporated this information into the biplot to elucidate the interconnections among variables and their significance (Figure 3). The details of the obtained principal components are presented in Table 3. Statistical analysis was performed using R (version 4.3.1) with packages (corrr [20], FactoMineR [21], factoextra [22]), Python (version 3.11.4) with packages (NumPy [23], Pandas [24], matplotlib [25], seaborn [26], and Statistica (version 13.3.721.1).

#### 2.4.1. Machine Learning Model

There is a lack of theories describing which regression model will be the best in each situation; thus, we decided to create more than one model and choose the best one based on the mean absolute percentage error (MAPE). The following regression models were evaluated: soft vector regression (SVR), random forest regression (RFR), k-nearest neighbor regression (kNN regression), ridge regression, and segmented regression (SR). Each model was developed in a Python environment using the scikit-learn package [27].

#### 2.4.2. Model Evaluation

The dataset was divided into two parts—a training dataset and a test dataset—with each containing 90% and 10% of the original dataset, respectively (the splitting was done randomly and only once for all datasets). Moreover, a cross-validation technique with 5 folds was employed to address the overfitting problem. The hyperparameters for each model were chosen based on the results of the GridSearchCV function. Subsequently, the model with the obtained hyperparameters was trained on the training dataset and then evaluated on the test dataset using MAPE. As the model is intended for predicting RMR in real cases, the MAPE was calculated based on non-standardized data. Each model was compared based on the resulting MAPE, and the one with the least MAPE was selected. Additionally, a Bland–Altman plot was generated for each model to check for similar trends or biases between methods.

## 3. Results

Based on the body composition details, we observed that the majority of the study population had overweight or obesity, particularly visceral obesity. The mean mass of body fat constituted almost 35% of the total body weight, indicating a substantial amount of body fat. The detailed anthropometric characteristics and body composition analysis of the study groups at baseline are presented in Table 4. In the study group, the mean BMI was 32.1 ± 8.5 kg/m^2^. Of the study population, 74.8% (89 individuals) were classified as overweight/obese. The majority of women (57.1%, *n* = 68) fell into the obesity category, defined by a BMI ≥ 30 kg/m^2^. The distribution of BMI in the study population is illustrated in Figure 4.

### 3.1. Soft Vector Regression (SVR)

Hyperparameters were determined through the utilization of GridSearchCV employing a fivefold cross-validation approach: C = 1, kernel = “poly,” degree = 1. The mean absolute percentage error (MAPE), computed on the non-standardized test dataset, was found to be 0.1189 (11.89%). The Bland–Altman plot revealed a tendency of the algorithm to underestimate for lower values and overestimate for the highest values of resting metabolic rate (RMR), as depicted in Figure 5.

### 3.2. Random Forest Regression (RFR)

Hyperparameters were determined after utilizing GridSearchCV with fivefold cross-validation: bootstrap = true, “max_depth” = 10, “max_features” = “sqrt”, “min_samples_leaf” = 4, “min_samples_split” = 2, “n_estimators” = 200. The mean absolute percentage error (MAPE), calculated on the non-standardized test dataset, was found to be 0.1424 (14.24%). The Bland–Altman plot (Figure 6) reveals a nearly negligible tendency of the algorithm to underestimate for lower values and overestimate for the highest value of resting metabolic rate (RMR). This observation may be attributed to the construction of the RFR algorithm, which is based on the bagging method.

### 3.3. k-Nearest Neighbor (kNN) Regression

Hyperparameters were determined after employing GridSearchCV with fivefold cross-validation: “algorithm” = “auto”, “n_neighbors” = 6, “weights” = “uniform.” The mean absolute percentage error (MAPE), computed on the non-standardized test dataset, was found to be 0.1592 (15.92%). The Bland–Altman plot (Figure 7) indicates a tendency of the algorithm to underestimate for lower values and overestimate for the highest value of resting metabolic rate (RMR).

### 3.4. Ridge Regression

Hyperparameters were determined following the application of GridSearchCV with fivefold cross-validation: “alpha” = 0.0001, “solver” = “auto”, “tol” = 0.001. The mean absolute percentage error (MAPE), assessed on the non-standardized test dataset, amounted to 0.1207 (12.07%). The Bland–Altman plot (Figure 8) illustrates a tendency of the algorithm to underestimate for lower values and overestimate for the highest value of resting metabolic rate (RMR).

### 3.5. Segmented Regression (SR)

Due to the limited size of the dataset, the decision on how to split it was predetermined, selecting the median (−0.0567) of RMR results. Subsequently, both datasets were partitioned into training and test sets using the previously described criteria. For each part, the support vector regression (SVR) method was applied with the following hyperparameters:

Part 1: C = 1.5, degree = 1, kernel = “poly”

Part 2: C = 1, degree = 1, kernel = “poly”

It is noteworthy that the splitting point was determined based on the measured results within our dataset. In typical scenarios where prior knowledge of RMR is unavailable, a defined methodology for splitting must be established. To address this, we propose the introduction of a system of equations for RMR calculation, as calculating RMR formulas independently may lead to overlapping results for a portion of the data.
RMRst = 0.04720 × PC1st + 0.0452 × PC2st + 0.0509 × PC3st − 0.600if0.0482 × PC1st + 0.0452 × PC2st + 0.0509 × PC3st − 0.600 ≤ −0.0567RMRst = 0.2160 × PC1st + 0.2184 × PC2st + 0.2116 × PC3st + 0.4945 otherwise

Certainly, if we want to convert the equation to the original version and denote σy as the standard deviation of y in the original dataset and xy as the mean of y, the original RMR can be obtained from the following equation:RMR = (0.0472 × PC1st + 0.0452 × PC2st + 0.0509 × PC3st − 0.6000) × σRMR + xRMRif0.0482 × PC1st + 0.0452 × PC2st + 0.0509 × PC3st − 0.6000 ≤ −0.0567RMR = (0.2160 × PC1st + 0.2184 × PC2st + 0.2116 × PC3st + 0.4945) × σRMR + xRMR otherwise

The next step is to outline the process for obtaining standardized principal components. By referring to the results presented in Table 3, one can derive the following:PC1 = 0.2328 × Age − 0.2903 × height + 0.2936 × weight + 0.3674 × BMI +0.2726 × LBM + 0.3374 × PBF + 0.3294 × MBF + 0.2066 × TBW + 0.3174 × VFL + 0.2962 × waist + 0.2795 × hips + 0.1853 × WHR
PC2 = 0.4479 × age + 0.4934 × height − 0.2250 × weight − 0.0695 × BMI − 0.2927 × LBM + 0.0497 × PBF − 0.1349 × MBF − 0.3635 × TBW + 0.0776 × VFL + 0.0535 × waist − 0.3113 × hips + 0.3951 × WHR
PC3 = 0.3150 × age − 0.1447 × height − 0.0650 × weight − 0.0215 × BMI − 0.1423 × LBM + 0.1300 × PBF − 0.0030 × MBF − 0.2148 × TBW + 0.3330 × VFL − 0.3125 × waist + 0.2579 × hips − 0.7189 × WHR

Therefore, the calculation of standardized principal components proceeds as follows:PC1st = [0.2328 × age − 0.2903 × height + 0.2936 × weight + 0.3674 × BMI + 0.2726 × LBM + 0.3374 × PBF + 0.3294 × MBF + 0.2066 × TBW + 0.3174 × VFL + 0.2962 × waist + 0.2795 × hips + 0.1853 × WHR − x_PC1_]/σ_PC1_
PC2st = [0.4479 × age + 0.4934 × height − 0.2250 × weight − 0.0695 × BMI − 0.2927 × LBM + 0.0497 × PBF − 0.1349 × MBF − 0.3635 × TBW + 0.0776 × VFL + 0.0535 × waist − 0.3113 × hips + 0.3951 × WHR − x_PC2_]/σ_PC2_
PC3st = [0.3150 × age − 0.1447 × height − 0.0650 × weight − 0.0215 × BMI − 0.1423 × LBM + 0.1300 × PBF − 0.0030 × MBF − 0.2148 × TBW + 0.3330 × VFL − 0.3125 × waist + 0.2579 × hips − 0.7189 × WHR − x_PC3_]/σ_PC3_

Table 5 presents the means and standard deviations needed for the above systems of equations.

The final equation is given by:RMR = (0.0472 × PC1st + 0.0452 × PC2st + 0.0509 × PC3st − 0.600) × 310.5558 + 1693.5234if0.0482 × PC1st + 0.0452 × PC2st + 0.0509 × PC3st − 0.600 ≤ −0.0567RMR = (0.2160 × PC1st + 0.2184 × PC2st + 0.2116 × PC3st + 0.4945) × 310.5558 + 1693.5234 otherwise

The particular variables can be obtained from the following equations:PC1st = [0.2328 × age − 0.2903 × height + 0.2936 × weight + 0.3674 × BMI + 0.2726 × LBM + 0.3374 × PBF + 0.3294 × MBF + 0.2066 × TBW + 0.3174 × VFL + 0.2962 × waist + 0.2795 × hips + 0.1853 × WHR − 109.5763]/34.5448
PC2st = [0.4479 × age + 0.4934 × height − 0.2250 × weight − 0.0695 × BMI − 0.2927 × LBM + 0.0497 × PBF − 0.1349 × MBF − 0.3635 × TBW + 0.0776 × VFL + 0.0535 × waist − 0.3113 × hips + 0.3951 × WHR − 109.9039]/34.5661
PC3st = [0.3150 × age − 0.1447 × height − 0.0650 × weight − 0.0215 × BMI − 0.1423 × LBM + 0.1300 × PBF − 0.0030 × MBF − 0.2148 × TBW + 0.3330 × VFL − 0.3125 × waist + 0.2579 × hips − 0.7189 × WHR − 108.9802]/34.5062

The aforementioned system of equations obviates the necessity for prior knowledge of RMR and yields consistent results. Employing this equation produces an MAPE of 10.78, representing the optimal outcome among all models. Furthermore, it is noteworthy that the Bland–Altman plots exhibit a similar trend to that observed in the previous case, albeit on a reduced scale (Figure 9).

## 4. Discussion

This study aimed to develop a predictive equation for resting metabolic rate (RMR) specifically tailored to women with lipedema, a chronic progressive disease characterized by abnormal fat accumulation. The importance of accurately estimating RMR lies in its role in determining overall energy expenditure, which is crucial for managing weight, especially in conditions like lipedema, where weight gain is linked to disease progression [8,9]. Our recent study identified a low-carbohydrate, high-fat diet with anti-inflammatory properties as the most effective treatment [28]. However, the precise calculation of resting metabolic rate and the energy value of the diet remains unknown. In clinical practice, the Mifflin et al. [29] equation is commonly applied, specifically tailored for overweight/obese patients, while the Harris–Benedict [30] equation is used for patients with normal body weight. However, our previous study demonstrated that these equations lack accuracy for females with lipedema, showing significant individual differences when compared to the actual resting metabolic rate measured by indirect calorimetry. Consequently, we have concluded that the development of a new equation dedicated to this patient group is necessary due to the inaccessibility of indirect calorimetry in hospitals and outpatient clinics [13]. Lipedema patients exhibit distinct differences from individuals with overweight or alimentary obesity, resulting in unique nutritional requirements. In comparison to individuals with obesity, lipedema patients show an increased adipocyte area. The number of macrophages exhibits a significant elevation in both the skin and fat of the lipedema group compared to those with obesity. Furthermore, there is an observed augmentation in dermal vessels in the non-obese lipedema group when contrasted with the obese group [31]. Additionally, particular authors strongly advocate for the development of new resting metabolic rate prediction equations that extend beyond the Caucasian race, considering different anthropometric characteristics, genetic backgrounds, lifestyle factors, and nutritional habits in diverse populations [10].

The newly created and validated equation for predicting resting metabolic rate in individuals with lipedema addresses a significant gap in accurate yet straightforward RMR prediction techniques. This new equation offers an easy and widely applicable alternative to indirect calorimetry, which is both expensive and not widely accessible. The robust correlation observed between the RMR measured through indirect calorimetry and the RMR predicted by the equation developed in this study enhances the practical utility of our new equation.

The current study signifies the development of an RMR prediction equation that incorporates body composition parameters, including lean body mass, body fat mass, total body water, and visceral fat level. Acknowledging the disproportion between the upper and lower parts of the body, a typical characteristic of lipedema patients, we have also incorporated waist, hips, and waist-to-hip ratio (WHR) to enhance the accuracy of the new equation. This is in addition to including height, weight, body mass index (BMI), and age in the typical predictive model [29,30]. Kfir et al. [32] emphasized the significance of incorporating body composition elements into the RMR equation. In their study, a newly proposed model incorporated sex, age, fat mass (FM), and fat-free mass (FFM). Their research highlighted a substantial disparity between commonly used prediction equations and measured RMR, advocating for the development of a more accurate equation that incorporates both fat mass (FM) and fat-free mass (FFM). The findings from the study conducted by Almajwal et al. [12] further substantiate the observation that incorporating body composition variables such as TBW (total body water), FFM (fat-free mass), and FM (fat mass) is valuable in the development of a new equation. Oliveira et al. [33] also emphasized the significance of incorporating various variables, including weight, height, FFM (fat-free mass), FM (fat mass), and AC (abdominal circumference), in their new equation designed for the severely obese female population.

Our findings align with the study conducted by Thom et al. [34], which reported a correlation between increasing age, height, and BMI with the underestimation of RMR. Additionally, they observed that the overall accuracy of the equations for predicting RMR was limited at the individual level, especially among women with both low and high RMR. Importantly, the technique used in our study significantly reduced the overestimation of RMR, highlighting its substantial value in improving accuracy. Flack et al. [35] observed the accuracy of all tested equations (including the Harris–Benedict and Mifflin equations) declined as fat-free mass (FFM) increased. This study supports the idea that individual variations in FFM contribute to variability in both resting metabolic rate (RMR) and RMR prediction accuracy among individuals. Model evaluation based on the mean absolute percentage error (MAPE) revealed that the segmented regression model outperformed the others, with an MAPE of 10.78%. The segmentation approach considered a split point based on the median RMR value, addressing potential overlapping results in real-case scenarios. The resulting equation, combining principal components and specific conditions for different RMR calculation formulas, demonstrated consistency and accuracy.

RMR predictions are crucial for tailoring effective treatment plans, including personalized dietary recommendations and exercise regimens, for women with lipedema. The proposed predictive equation, considering the unique metabolic characteristics of lipedema patients, provides a valuable tool in clinical practice, where indirect calorimetry may be inaccessible. Moreover, the study acknowledges the limitations, such as the necessity of prior knowledge for splitting datasets in the segmented regression model. Despite this limitation, the consistent performance of the proposed equation and its alignment with Bland–Altman plots support its reliability.

## 5. Conclusions

In conclusion, this study contributes to the advancement of personalized care for women with lipedema by offering a reliable predictive equation for RMR. The integration of machine learning techniques and segmentation approaches addresses the challenges posed by limited sample size and complex relationships among variables. Future research may explore the application of these findings in larger cohorts and diverse populations, further enhancing the understanding and management of lipedema.

## Figures and Tables

**Figure 1 metabolites-14-00235-f001:**
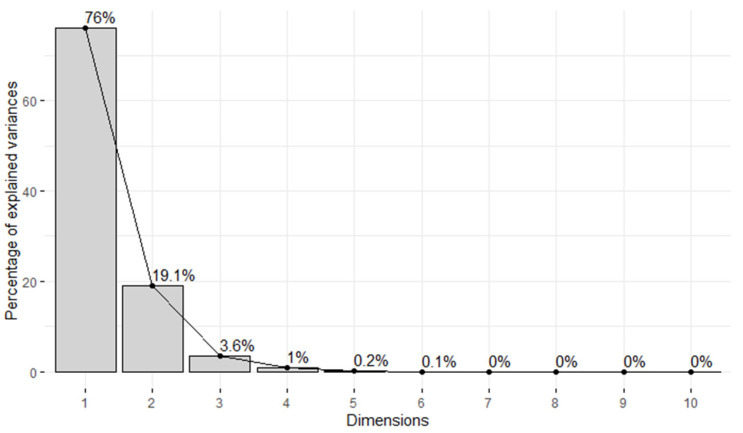
Scree plot for PCA.

**Figure 2 metabolites-14-00235-f002:**
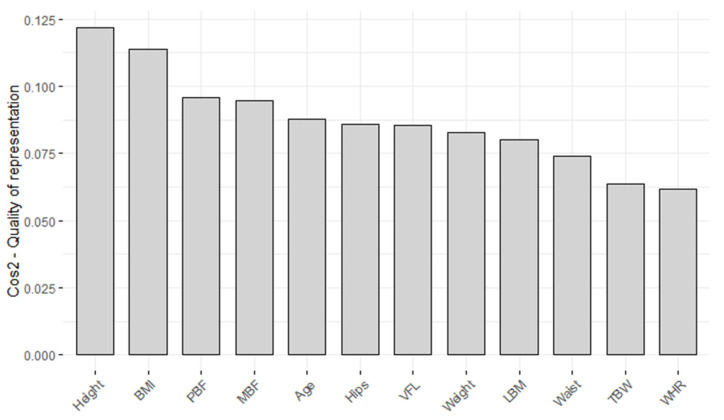
Cos2 of variables to Dim-1-2.

**Figure 3 metabolites-14-00235-f003:**
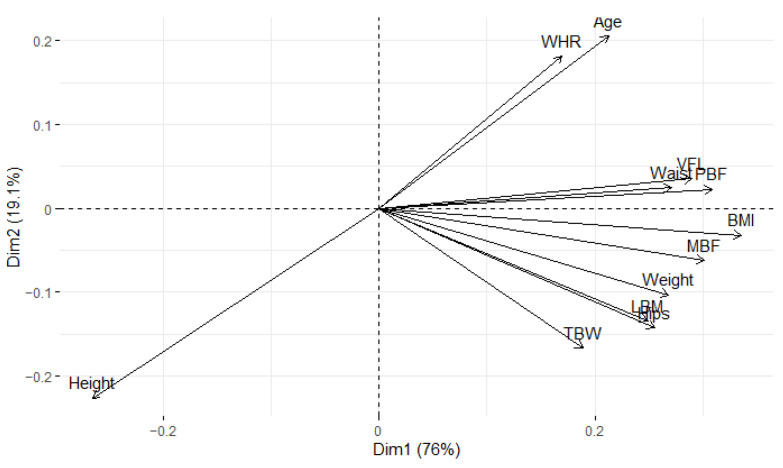
Projection on the first two principal components with presented results for quality of representation from the Figure 2.

**Figure 4 metabolites-14-00235-f004:**
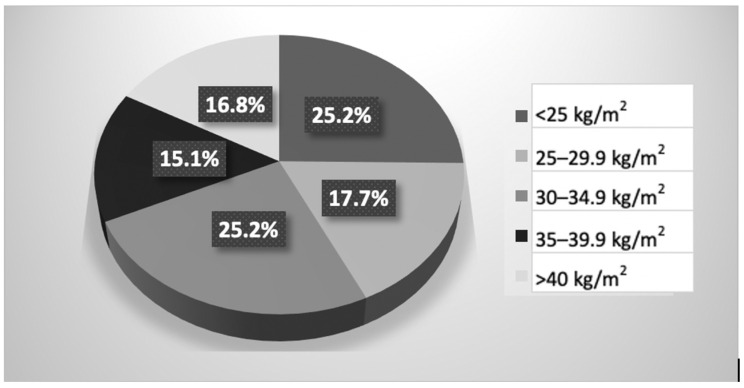
Prevalence of BMI categories in the study group.

**Figure 5 metabolites-14-00235-f005:**
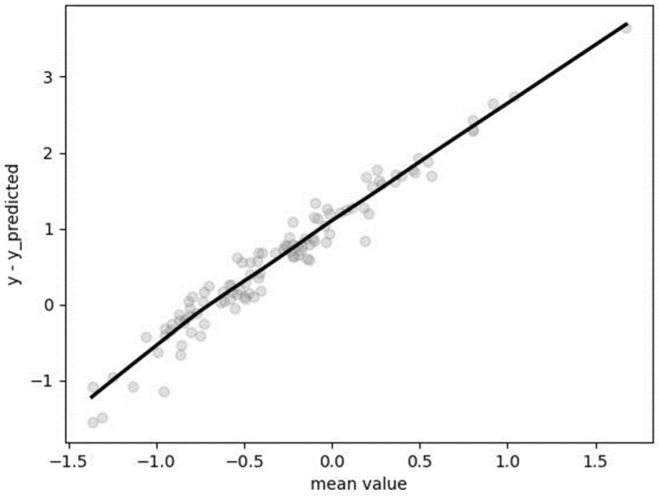
Bland–Altman plot for results obtained by SVR.

**Figure 6 metabolites-14-00235-f006:**
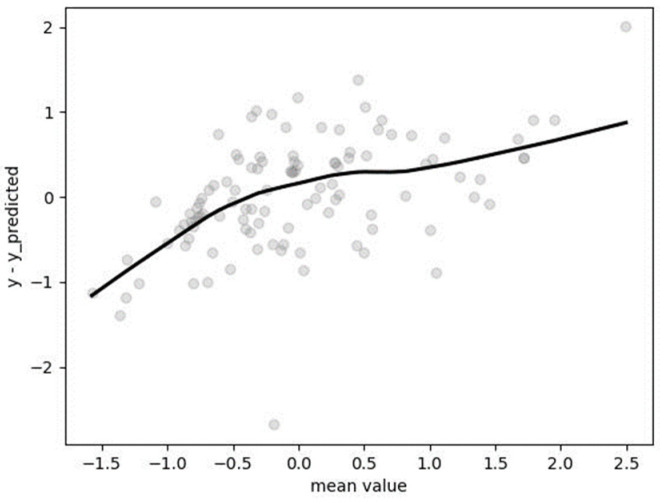
Bland–Altman plot for results obtained by RFR.

**Figure 7 metabolites-14-00235-f007:**
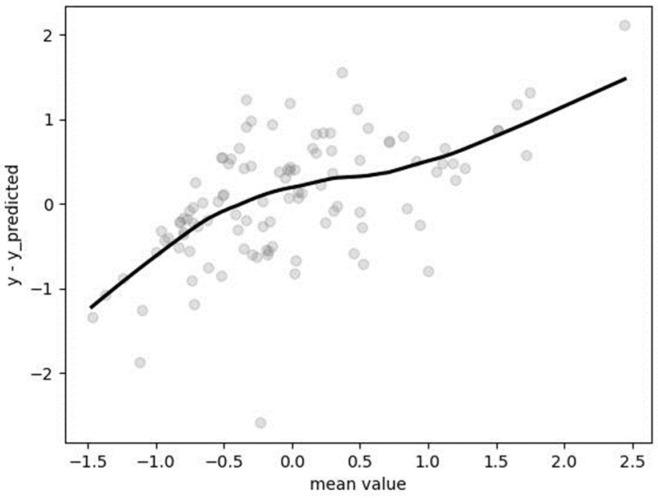
Bland–Altman plot for results obtained by kNN regression.

**Figure 8 metabolites-14-00235-f008:**
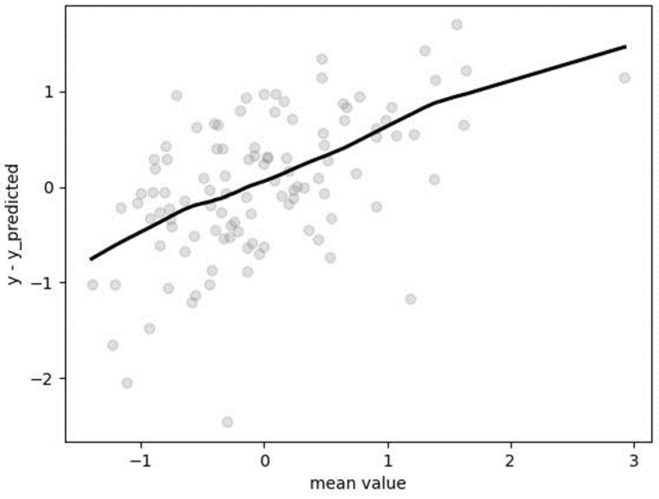
Bland–Altman plot for results obtained by ridge regression.

**Figure 9 metabolites-14-00235-f009:**
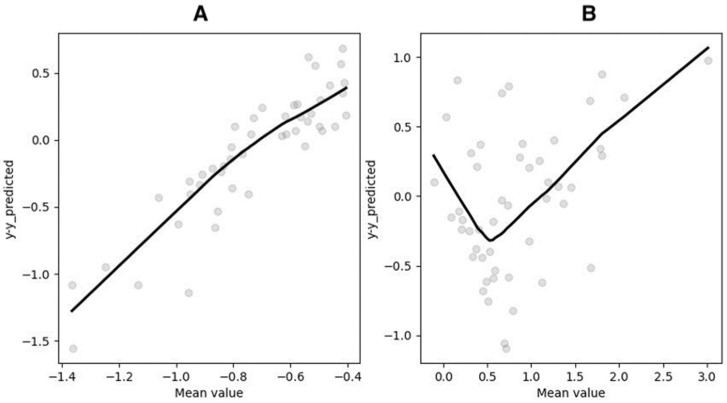
Bland–Altman plots for results obtained by SR divided into two parts: (**A**) first part of dataset, (**B**) second part of dataset.

**Table 1 metabolites-14-00235-t001:** Differences between lipedema and lymphedema.

Characteristic	Lipedema	Lymphedema
Gender	Female	Female and male
Onset	Puberty, pregnancy, menopause	Childhood to elderly
Family history	Common	In some cases of primary lymphedema
Areas affected	Buttock, hips, legs, sometimes arms	Legs and feet, arms and hands
Symmetry	Often	Possible
Pain in the legs/arms	Often	Rare
Tenderness of legs/arms	Often	Rare
Easy bruising	Often	Absent
Pitting edema	Absent	Present
Affected feet	Absent	Present
Response to diet and exercises	Absent	Possible

**Table 2 metabolites-14-00235-t002:** Variance inflation factor for all predictors.

Age	Height	Weight	BMI	LBM	PBF	MBF	TBW	VFL	Waist	Hips	WHR
2.74	40.64	864.54	375.52	106.58	26.57	240.12	37.01	12.00	197.63	96.05	88.62

BMI, body mass index; LBM, lean body mass; PBF, percentage body fat; MBF, mass body fat; TBW, total body water; VFL, visceral fat level; WHR, waist-hip ratio.

**Table 3 metabolites-14-00235-t003:** Loadings of the first three principal components.

Parameter	PC 1	PC 2	PC 3
Age_st_	0.2328	0.4479	0.3150
Height_st_	−0.2903	0.4934	−0.1447
Weight_st_	0.2936	−0.2250	−0.0650
BMI_st_	0.3674	−0.0695	−0.0215
LBM_st_	0.2726	−0.2927	−0.1423
PBF_st_	0.3374	0.0497	0.1300
MBF_st_	0.3294	−0.1349	−0.0030
TBW_st_	0.2066	−0.3635	−0.2148
VFL_st_	0.3174	0.0776	0.3330
Waist_st_	0.2962	0.0535	−0.3125
Hips_st_	0.2795	−0.3113	0.2579
WHR_st_	0.1853	0.3951	−0.7189

PC, principal component; BMI, body mass index; LBM, lean body mass; PBF, percentage body fat; MBF, mass of body fat; TBW, total body water; VFL, visceral fat level; WHR, waist-hip ratio. We denoted the standardized variable “st” as a lower index.

**Table 4 metabolites-14-00235-t004:** Clinical characteristics of the study groups at baseline.

Parameter	Lipedema, *n* = 119 Mean ± SD
Age (years)	43.4 ± 13.4
Height (cm)	165.5 ± 6.8
Weight (kg)	87.5 ± 21.8
BMI (kg/m^2^)	32.1 ± 8.5
LBM (kg)	48.7 ± 19.8
PBF (%)	37.7 ± 7.3
MBF (kg)	34.4 ± 13.8
TBW (kg)	38.6 ± 6.5
VFL	12.7 ± 5.1
Waist (cm)	96.5 ± 17.5
Hips (cm)	115.6 ± 13.6
WHR	0.8 ± 0.1
RMR (kcal/day)	1685.8 ± 310.4

BMI, body mass index; LBM, lean body mass; PBF, percentage body fat; MBF, mass body fat; TBW, total body water; VFL, visceral fat level; WHR, waist-hip ratio; RMR, resting metabolic rate.

**Table 5 metabolites-14-00235-t005:** Mean and standard deviation of RMR and first three principal components.

Parameter	Mean	SD
PC1	109.5763	34.5448
PC2	109.9039	34.5661
PC3	108.9802	34.5062
RMR	1693.5234	310.5558

## Data Availability

All data used to support the findings of this study are available from the corresponding author upon reasonable request due to privacy of patients.

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
