# Peer review of "Development of New Predictive Equations for the Resting Metabolic Rate (RMR) of Women with Lipedema"

_metabolites, 2024, doi:10.3390/metabo14040235_

Round 1

Reviewer 1 Report

Comments and Suggestions for Authors

Lipedema is a chronic condition characterized by the abnormal buildup of adipose tissue in certain parts of the body, most commonly the lower extremities, resulting in bodily deformity, pain, reduced motor function, and so on. This condition is unique in that it resists physical exercise therapy. This condition is unique in that it resists physical activity therapy. Frequently, altering the calorie composition of the diet while not exceeding the

Because there is no consistent methodology for determining resting metabolic rate, the work provided here is well-structured, clear, and useful.
More over half of the citations are less than 5 years old, and no evidence of excessive self-citation was found.

The authors justify the need for new mathematical calculations to estimate the resting metabolic rate in lipedema and propose an algorithm based on a combination of regression and clustering methods (support vector method, Random Forest Regressor, k-nearest neighbors algorithm, cross-validation, Bland-Altman plot, variance inflation factor, and principal component analysis).
It is possible to repeat this study if the research team includes specialists in mathematical modeling and statistical analysis, as well as the software packages used in this work and a sufficient number of lipedema patients, but the likelihood of inconsistencies in the results is high due to ethnic reasons.

Readers inexperienced with such quantitative analytical methods will find the graphical drawings offered difficult to understand.
In theory, the authors propose a presonified approach for assessing resting metabolic rate in lipedema patients, which is widely used in most countries.
The study was carried out in compliance with the Declaration of Helsinki and ethical committee decision. 

Author Response

Dear Reviewer,

We sincerely appreciate your recognition of our study and its obtained results. Your acknowledgment of the well-structured, clear, and useful nature of our study is truly gratifying. We understand that readers unfamiliar with quantitative analytical methods may find the graphical representations challenging to interpret. Nevertheless, we remain hopeful that our methods will spark interest among fellow researchers, who may also appreciate the efforts we have invested.

Thank you once again for your valuable feedback.

Sincerely,

Małgorzata Jeziorek

Reviewer 2 Report

Comments and Suggestions for Authors

Article «Development of new predictive equations for the resting metabolic rate (RMR) of women with lipedema» (Authors: Małgorzata Jeziorek and at. all). The article is written in a clear and understandable language, the literature corresponds to the stated topic. The work is devoted to the development of an improved predictive calorie accounting model used for nutrition planning for patients with lipedema. The aim of the study was to derive a predictive equation for resting metabolic rate (RMR) using a sample of 119 women with lipedema.

Some comments.

1.     Please add a confidence interval in Figure 2.

2.     Machine learning requires a significant database. More than 119 patients. Nevertheless, the current result will be useful for further development of this direction.

 However, I would recommend publishing the work.

Author Response

Dear Reviewer,

Thank you for your review, we sincerely appreciate your recognition of our study and your recommendation for publication.

Thank you for your suggestion regarding the inclusion of confidence intervals in Figure 2. Generally, graphs depicting the contribution of each variable to the principal component (defined as cosine squared (Cos2)) do not incorporate confidence intervals. This practice aligns with common conventions observed in publications and literature. As an example, I would like to direct your attention to chapter 12.8 of the following resource: https://biosakshat.github.io/pca.html#quality-of-representation.

We understand the significance of having a substantial database for machine learning methods. Your comment regarding the importance of a dataset comprising more than 119 patients for robust analysis and generalization of results is important for us. We are fully committed to furthering our work in this direction. While we acknowledge this limitation, we are encouraged by your recognition that our current findings remain relevant for advancing this area of research.

Sincerely,

Małgorzata Jeziorek

Reviewer 3 Report

Comments and Suggestions for Authors

Congratulations very good manuscript. It is certainly very specialized for experts in the field of lipedema. You would try to semplify the results and insert some elements that allow the readers to differenziate lipedema from lymphoedema..

Author Response

Dear Reviewer,

Thank you for your review, we sincerely appreciate your recognition of our study.

We would like to simplify the results as much as possible so that they are easier to understand for the reader. However, this would mean removing some of the relevant results. We believe that our method of presenting the results is as clear and transparent as we could presented it.

In response to your suggestion, we have integrated information distinguishing lipedema from lymphedema into a new Table 1, which is now included in the Introduction section. We believe this addition will aid readers in differentiating between the two conditions.

Sincerely,

Małgorzata Jeziorek
